# A Visual User Interfaces for Constant Checking of Non-Invasive Physiological Parameters

**Sara Jelbeb and Ahmad Alzubi ***

Business Administration Department, University of Mediterranean Karpasia, Institute of Graduate Research and Studies, TRNC, 33010, Mersin 10, Turkey; sara.melad89@gmail.com
***** Correspondence: ahmad.alzaubi@akun.edu.tr

**Abstract:** Objective: this study proposes the development of a wireless graphical interface with a monitoring system that allows for extensive integration with a variety of non-invasive devices. Method: an evaluation framework was created using ISO/IEC25012 parameters to evaluate each of the physiological parameters. Using an ISO standard as a framework to evaluate the quality of the results and analysis parameters such as consistency, accessibility, compressibility, and others, the Cayenne myDevices platform is used to develop a variety of IoT projects. Results: the successful prototype shows that the temperature sensor's technical capabilities were found to be insufficient for accurately measuring a human's body temperature, requiring a calibration algorithm. The Cayenne myDevices platform provides a web dashboard for continuous tracking and storage of physiological data. Blynk, an IoT-based application with a graphical user interface, enables real-time visualization and tracking of data from the server and the electronic prototype. Conclusion: findings concluded that free software tools such as Cayenne myDevices, the Blynk App, and Arduino enable integration and reduce the need for expensive applications. Electronic prototypes monitor parameters (e.g., temperature, heart rate, oxygen saturation) were used to monitor COVID-19, cardiovascular, and diabetic patients during exercise. Successful prototypes used Max30100, Mlx90614 sensors, and Esp8266 microcontroller. To avoid giving the patient inaccurate results, the instruments must be carefully selected, so they were assessed to ensure a 95% effectiveness level.

**Keywords:** physiological parameters; IoMT; prototype; portable sensors; Cayenne myDevices platform; Blynk app

## 1. Introduction

One of the most popular technologies is the Internet of Things (IoT), which is referred to as a global network that includes a monitoring system that gathers data thanks to IoT sensors and enables the processing and analysis of that data [1]. IoT devices were created and can adopt protocols that are used depending on the needs or requirements they provide; devices can also connect with each other. Today, humanity depends on the Internet, and to obtain true and high-performance communication, IoT devices were developed. To enable the exchange of information on a network from anywhere in the world, in addition to being able to send and receive data between electronic devices connected to a Wi-Fi network, communication protocols are made up of layers that provide specific functionalities that make up a set of rules [2].

As IoT researchers build more dependable, adaptable networks, the number of mobile devices connected to the Internet increases over time. Up until 2020, there were at least 8 billion devices connected to the Internet without counting phones [3]. Deep neural network-powered IoT in healthcare has resulted in ground-breaking advancements for the medical community, distinctive probability, and medical data analysis in the healthcare sector [4]. In order to do this, monitoring systems have been put in place that support human health and medical care while utilizing IoT technology to support ongoing and remote monitoring.

Its structures are crucial because they increase access to immediate medical care, cut costs associated with travel to and from medical appointments, and generally improve health [5]. Hospitals have also created sensors that use deep learning methods, such as seizure neural networks, to measure a variety of parameters, including temperature, heart rate, and blood pressure [6].

The study by Rodrguez et al. in 2019 [7] sought to design a chair for everyday use by incorporating wireless sensors to monitor air quality and heart rate continuously in real-time; they employed a variety of hardware and open-source software tools, including PPG sensors (Photoplethysmography), a simple and low-cost optical technique used to calculate heart rate, and ECGs (Electrocardiogram), which are used to extract signals corresponding to heart rate variability. A Raspberry Pi for data transmission and a web app for information visualization were used in this study. The device was dependable and trustworthy, producing excellent results crucial for those with high or low blood pressure.

Patil et al., 2019 [8] created an erratic electronic belt to continuously monitor the body's pulse and body temperature. As hardware, it used a Wemo microcontroller, which is smaller than the Arduino but performs the same functions. The software incorporates the same sensors as the first study, with the exception that it uses GMSs.

Murali and Swathi, 2020 [9] designed an IoT-based pulse oximeter to track specific physiological parameters, including SpO2, heart rate frequency, and blood pressure. To do this, they used GPS software to determine the precise coordinates of the patient and some sensors already mentioned in the second study [8], such as GMS and PPG. With the aid of a new sensor built into this prototype, the GPRS (General Packet Radio Service), which connects with the GMS to generate an alert, calls the emergency room, and sends a text message to the patient's nearest family members, it was possible to create a mobile application.

Among the various types of biosensors, electrochemical (EC), optical, and magnetic biosensors have been successfully developed for the detection of tumor biomarkers, including those related to prostate cancer, in clinical and point-of-care (POC) settings. While POC biosensors have shown potential for detecting PC biomarkers, certain limitations, such as sample preparation, need to be addressed. To overcome these challenges, researchers have employed new technologies to develop more practical and efficient biosensors, offering promising prospects for their integration into clinical practice [10].

Chen et al., 2022 [11] have explored various respiration monitoring devices, including wearable sensors integrated into clothing and smartphone-based approaches. While these methods offer convenience, challenges persist in achieving accurate and user-friendly designs for continuous monitoring. This work presents a soft, wireless, and non-invasive device for real-time evaluation of human respiration, addressing the need for effective portable electronic devices in medical and daily respiration monitoring. The device offers high accuracy and usability, promising improved respiratory health monitoring and medical assistance.

An et al., 2019 [12] have presented microfluidic contact lens sensors for unpowered, continuous, and non-invasive intraocular pressure (IOP) monitoring. The sensors consist of a soft-elastomer sensing layer and a hard plastic reference layer, with an annular sensing chamber filled with dyed liquid and a sensing microchannel serving as the IOP transducer. The IOP signal is detected from the displacement change of the dyed liquid's interface in the sensing channel, which can be observed optically using a smartphone camera. The sensors demonstrate high sensitivity (0.708 mm/mmHg), good reversibility and long-term stability. Tests on porcine eyes ex vivo show promising results, with a sensitivity of 0.2832 mm/mmHg in the range of 8~32 mmHg and good reproducibility. This work represents a significant advancement in IOP monitoring for glaucoma management.

Cao et al., 2021 [13] aimed to improve the performance of non-invasive EEG-based brain-computer interfaces (BCIs) for robotic arm control, where a proposed novel shared control model dynamically optimizes the control commands based on real-time context and user characteristics. Additionally, a hybrid BCI scheme is introduced, employing steady-state visual evoked potentials and motor imagery to enable multi-dimensional control of

the robotic arm. The results demonstrate significant improvements in the accuracy and success rate of completing tasks using the shared control strategy with hybrid BCI, making brain-actuated robotic arm systems more effective.

Birtek et al., 2023 [14] showed that machine learning models can optimize microfluidic systems by predicting outcomes prior to fabrication, minimizing cost and time. An educational interactive microfluidic module was developed for designing efficient micromixers at low Reynolds regimes for Newtonian and non-Newtonian fluids. The machine learning model achieved high accuracy (R2 = 0.9543) for predicting the mixing index and optimizing micromixer designs for Newtonian fluids. Non-Newtonian fluid cases were also optimized using machine learning (R2 = 0.9063). The framework serves as an interactive educational module, demonstrating the integration of artificial intelligence in engineering education.

This study aims to advance the creation of an IoT-based non-invasive electronic prototype for targeted physiological parameter monitoring. It also offers the incorporation of an Internet of Things (IoT)-based application to read patient data, with a graphical user interface that emphasizes the use of a widget provided by both the application and the server to be able to determine data such as blood saturation, heart rate, and temperature in order to deliver succinct information in the present and lower the likelihood of complications in the future. Since COVID-19 attacks the entire immune system and puts all countries in the world in a precarious situation, it has caused a number of complications for people with pre-existing diseases such as cardiovascular, asthmatic, and pulmonary. Devices have also been developed and put into use to address these issues, but they are expensive and not readily accessible to patients. Since the world is constantly undergoing technological change, it is crucial to create a device with a dependable graphical interface that is simple to use, compressible, has high-quality standards, and is capable of accurately interpreting what the interface shows, in this case, the patient's vital signs. This work aims to create a graphical user interface application to monitor physiological parameters in real-time using a non-invasive electronic prototype.

Overall, this research offers a comprehensive approach to creating a wireless monitoring system that combines IoT technology, non-invasive sensors, free software tools, and a user-friendly GUI application. By addressing real-world healthcare challenges and utilizing innovative sensor combinations, the study provides a valuable contribution to the field of remote physiological monitoring. The successful implementation of such a prototype could have significant implications for healthcare, enabling continuous monitoring and improving patient outcomes, especially for individuals with pre-existing medical conditions.

## 2. Materials and Methods

### 2.1. Delimitation

This work was carried out explicitly to evaluate and develop an electronic prototype in order to monitor physiological parameters in patients, from youth to older adults, which was carried out in the city of Tarablous, Libya, for which tests were developed. Laboratory experiments were prepared by its author. An estimated period of 9 months was established for its preparation, which began in June 2021 and ended in July 2022.

### 2.2. Types of Research

Based on what was proposed in this research work and in accordance with the objectives, it has been considered that this study is defined as qualitative, quantitative and laboratory. It was important to use this type of research because tests of the electronic prototype were carried out at our institution to observe and analyze the effectiveness of the data.

### 2.3. Research Method

The possible research methods will be defined, and how the process was carried out according to the type of research, which is why the inductive, deductive and experimental methods were used.

### 2.4. Population

The population for this research is specifically focused on the teachers who attended in person, in which 150 stratified sampling was used to find those between 20 and 69 years old. As a result, a sample of 27 teachers was achieved; Equation (1) presents the finite population formula applied for this research since the number of teachers who work face-to-face at our university is known.

The formula is as follows:

$$n = \frac{N \times Z^2 \times p \times q}{d^2 \times (N-1) + \left( Z_a{}^2 \times p \times q \right)} \tag{1}$$

where: N = population size (150); Z = confidence level (95%); $Z_a$ = Z-score corresponding to a level of confidence according to the standard normal distribution (for a level of confidence of 95%, z = 1.96; p = probability of success, or expected proportion (0.3); q = probability of failure (0.3); d = precision (Maximum permissible error in terms of proportion) (0.10).

### 2.5. Inclusion and Exclusion Criteria

Inclusion criteria:

- Teachers who were actively teaching and physically present at our university during the study period.
- Individuals aged between 20 and 69 years old.
- Participants willing to participate in the testing process of the electronic prototype and provide informed consent.

Exclusion criteria:

- Teachers who were on leave or not physically present at the university during the study period.
- Individuals under the age of 20 or over the age of 69.
- Teachers who declined to participate or were unable to provide informed consent for any reason.

These criteria were put in place to ensure the consistency of our data and the appropriateness of our test subjects for the objective of our study, which was to evaluate the electronic prototype across a broad age range of active teachers.

### 2.6. Investigation Techniques

The techniques used for this research were: interviews, surveys, scientific observation, and experiments. The first research technique used was an interview with an expert in the medical field, a nurse who works at our institution, in order to ascertain, from his point of view, the level of acceptance of the data provided by the electronic prototype. The second technique used in the investigation consisted of a structured survey addressed to the participants to ascertain the use and degree of satisfaction with the electronic prototype; to determine this, 10 closed questions were posed. In addition, the scientific observation technique used to observe the events that occurred when implementing the electronic prototype instruments was applied, thereby verifying if the required characteristics were met. The scientific observation was carried out in a structured way, allowing the evaluation of some aspects. Evaluative parameters were assigned to each of these aspects from 1 to 5, where: 1 is insufficient, 2 is regular, 3 is acceptable, 4 is good, and 5 is excellent.

An evaluation sheet was also used based on the ISO/IEC 25012 standard; among the characteristics, we have accuracy, precision, and availability, among others, to measure the veracity of the data provided by the electronic prototype.

On the other hand, at the time of evaluating the electronic prototype, a certain number of tests were carried out in conjunction with the sample presented above to obtain data collection from each of the subjects. An evaluation sheet for the electronic prototype was created in a structured way in which each of the parameters to be measured and the

aspects to be considered was evidenced; in addition, to improve the reliability of the data, a comparison of the data provided by the prototype was conducted. The following link contains the experimental data collection sheet for the electronic prototype. Additionally, to evaluate the quality of the electronic prototype, a structured form was designed; the aspects to be evaluated are ones that determine whether this prototype can meet a patient's needs in a certain way and have a significant value for patients in society.

### 2.7. Ethical Standards

The present study respects all the regulations established in order to comply with the rules decreed by the institution; confidentiality was respected during the collection of information such as bibliographic documents and books; likewise, free software licensing rights were used for the operation of the electronic prototype for the monitoring of physiological parameters. Ethical principles of health, such as respect for the person, Charity, Justice and Scientific Validity, were followed.

### 3. Results

### 3.1. Main Methods to Measure Physiological Parameters

This phase consisted of determining the sensors necessary for the investigation; the Mlx90614 sensor was applied, which measures body temperature without having to come into contact with the patient; in addition, the Max30100 sensor was used to detect SpO2 and BPM (Beats Per Minute); this is vital for the investigation to determine if the patient is in good health or has a problem.

### 3.2. Transmission Medium for Physiological Parameters

After comparing the microcontrollers mentioned above, it was concluded that the NodeMCU Esp8266 v3 was best, since it meets the necessary requirements for the development of the stated objectives and displays characteristics such as a compact design, connection in series between devices, compatibility with the sensors and easy connectivity with the Wi-Fi network, an essential data, in addition, it uses I2C communication which directs the sensors mentioned above.

### 3.3. Design and Elaboration of the Electronic Prototype

The diagram corresponding to the logical operation of the prototype was developed and implemented, as shown in Figure 1, where the prototype components are displayed and grouped into four logical blocks: physiological parameters, data collection, data transformation, and visualization of data. The first block corresponds to the physiological parameters and variables obtained when the prototype is used. The second block is based on obtaining data that comprises two modules that constitute the entire hardware part, being the Mlx90614 sensor for temperature and the Max30100 sensor to obtain heart rate and oxygenation. The third block is the NodeMCU Esp8266 v3 microcontroller in charge of Wi-Fi connectivity, processing the data and later sending the data that will be displayed within the graphical user interface and server. The fourth and last block is the visualization of data remotely; this is achieved thanks to the OLED screen and does not require Wi-Fi connectivity.

In order to avoid failures when carrying out the implementation of the prototype, a logic schematic was designed for the circuit.

Figure 2 details the operation of the prototype. The patient is the most important since it provides information on each of the measured physiological parameters; in this way, the sensors will obtain readings from the microcontroller, where they interact with each other with the help of the I2C protocol. The microcontroller executes a series of predefined actions in the source code, and at the same time, it sends data to the Cayenne server through the MQTT brokers, which are in charge of managing and controlling the IoT device so that it can be displayed and stored on the Cayenne dashboard. In addition, it contains alarm notifications previously configured on the server; in the same way, this information is sent to the Blynk application through a token that will obtain a connection directly from the

electronic prototype allowing the visualization of the parameters in the Blynk app GUI, so that real-time monitoring can be established both in the Blynk application and in the cloud server. A Wi-Fi connection is required; for this, it uses the TCP/IP communication protocol since it is integrated into the Esp8266 microcontroller and, thus, can access a network. In addition, it allows the electronic prototype to be anywhere with internet access, allowing the patient to check their physiological parameters remotely.

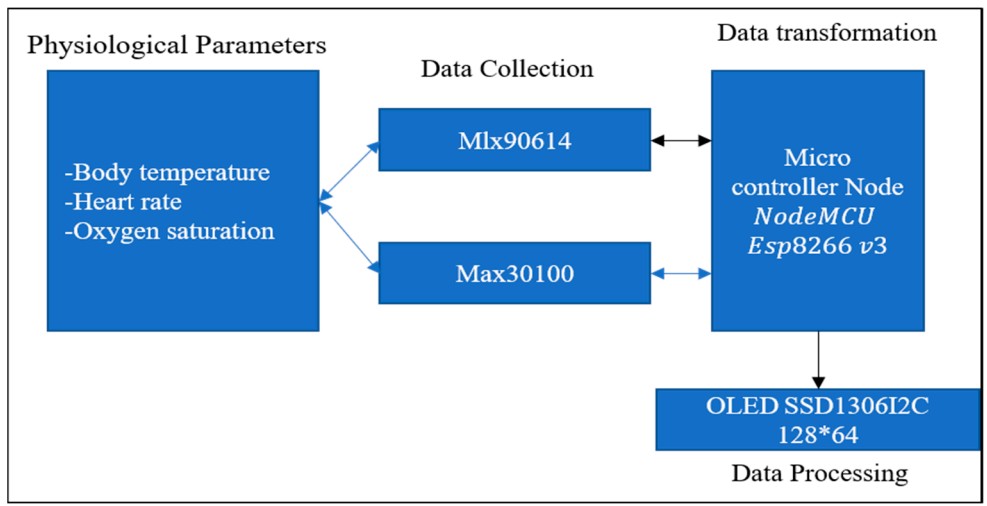

**Figure 1.** Block diagram of the electronic prototype.

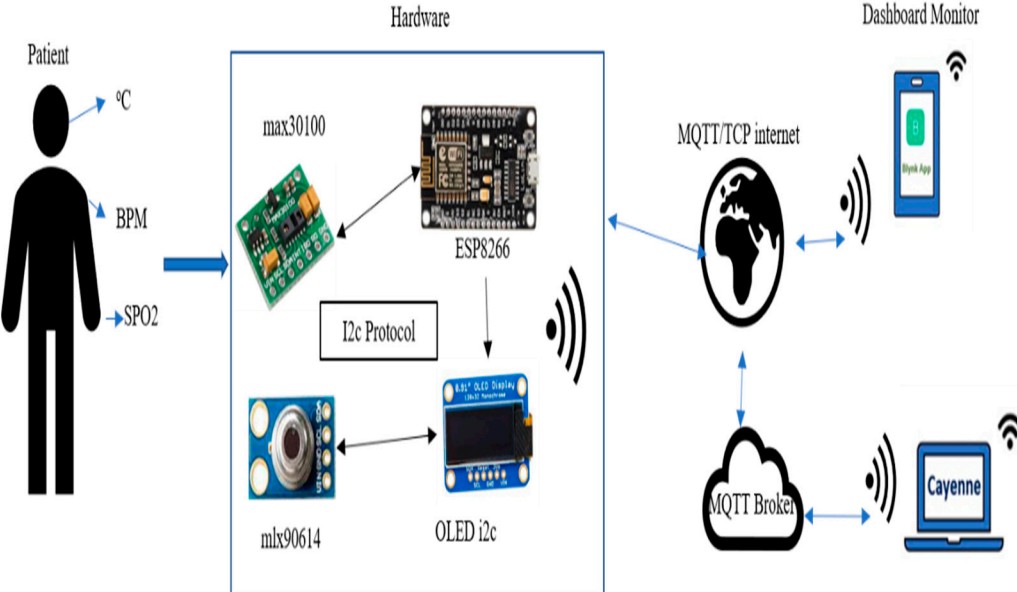

**Figure 2.** Logical operation of the electronic prototype.

The OLED is a display integrated into the electronic prototype itself. It provides a local display for the patient (user) to see their own physiological parameters in real-time. When the patient interacts with the electronic prototype, they can directly view their vital signs (e.g., body temperature, heart rate, oxygen saturation) on the OLED screen. The Blynk App serves as a graphical user interface (GUI) application on a mobile device (e.g., smartphone or tablet); it allows both the patient (user) and authorized healthcare professionals (such as doctors or caregivers) to remotely monitor the patient's physiological parameters. Finally, the Cayenne Dashboard is a cloud-based server that receives and stores data from the electronic prototype through MQTT brokers. It is primarily accessed and monitored by

healthcare professionals, researchers, or authorized personnel responsible for managing multiple patients' data. The Cayenne Dashboard provides a centralized platform for data visualization and analysis, enabling medical professionals to track patients' physiological parameters over time and detect any abnormalities or trends.

### 3.4. Implementation of the Prototype

The observation method used in this research was based on the most relevant parameters for each instrument to be used, these being the selected characteristics as Microcontroller (Connectivity; Memory; Compatibility; Availability) and Sensor (Feasibility; Security; Compatibility; Response Time).

A valuation table was implemented to better understand the results, as shown in Figure 3.

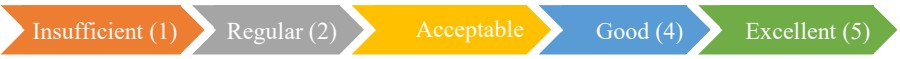

**Figure 3.** Scalar evaluation for the aspects to be evaluated.

Once the evaluation results have been presented based on the observation of each tool used, an explanation for the results shown can be established. As shown in Table 1, the section of the Esp8266 microcontroller was evaluated, which obtained results equivalent to 95% optimization. Connectivity was evaluated as the first parameter for a better understanding of the data, which obtained a score of five points. The information is sent quickly to the server and the app, allowing visualization of the data; however, it is important to point out that this may vary due to the speed of your home internet connection or wherever you are accessing the network from. Data Storage scored four points, given its ability to save data in its flash memory and allow the use of a considerable amount of memory. The third parameter, Compatibility, obtained a score of five points because of its high compatibility with the sensors and thanks to pro-I2C, SPI communication protocols, in addition to its 3.0 to 3.6 v power supply, thus allowing the use of all sensors that work in that range. The fourth and last parameter is availability, which obtained five points for being an easy-to-find microcontroller in Libya.

**Table 1.** Esp8266 Evaluation Results.

| Aspects to evaluate | Assessment | | | | |
| --- | --- | --- | --- | --- | --- |
| | 1 | 2 | 3 | 4 | 5 |
| connectivity | | | | | √ |
| Memory | | | | √ | |
| Compatibility | | | | | √ |
| Availability | | | | | √ |

Referring to Table 2, the following section was evaluated, which are the Mlx90614 and Max30100 sensors. They established a 95% optimization result in general; going into more detail, the temperature sensor Mlx90614 has been evaluated as follows: the parameter of Feasibility obtained a score of four points because despite showing great effectiveness in obtaining data, it generates small variations in degrees, correcting itself after a few readings, reducing the margin of error. Security is the second parameter, with a score of five points, since it guarantees the possibility of not having an accident related to electrical connections through the cover that the sensor has. The third parameter, Compatibility, obtained a score of five points for being highly consistent with the Esp8266 microcontroller; this is important because this way, we avoid having any problems in the future. The fourth parameter is Response time, which achieved a score of five points given that it is a sensor used in different areas and tends to read the data in less than a minute.

**Table 2.** Evaluation Results of Mlx90614 and Max30100 Sensor.

|  | Mlx90614 | | | | | Max30100 | | | | |
|---|---|---|---|---|---|---|---|---|---|---|
|  | Assessment | | | | | Assessment | | | | |
| Aspects to evaluate | 1 | 2 | 3 | 4 | 5 | 1 | 2 | 3 | 4 | 5 |
| Feasibility |  |  |  | √ |  |  |  |  | √ |  |
| Security |  |  |  |  | √ |  |  |  |  | √ |
| Compatibility |  |  |  |  | √ |  |  |  |  | √ |
| Response time |  |  |  |  | √ |  |  |  |  | √ |

Regarding the Max30100 heart rate sensor and oxygen saturation sensor, the same parameters described above were evaluated. The Feasibility parameter obtained a score of four points because the first readings, similarly to temperature, varied, showing low values that went up until reaching real values, reducing the margin of error; this is in accord with a medical oximeter, which does not show very low values. The second parameter is Safety, with a score of five points since it guarantees the possibility of not having an accident relating to connection; it could occur if the person has wet or sweaty hands, but even so, it is still practically imperceptible. The third parameter is Compatibility which obtained a score of five points for being largely compatible with the Esp8266 microcontroller. The fourth and last parameter is Response time, which achieved a score of five points due to or because it is a sensor used specifically in healthcare and tends to read data in less than a minute. Due to the effectiveness of the results, as mentioned above, the implementation of the prototype was managed and carried out in two phases. The first phase consisted of the individual test of each sensor; in order to be able to verify their use, in addition to verifying their operation. The implementation was carried out with the guidance of the schematic corresponding to each sensor. The first sensor to be implemented was the Mlx90614, corresponding to the temperature sensor, following the schematic as shown in Figure 4; in addition, you can see the use of this sensor in Figures 4 and 5.

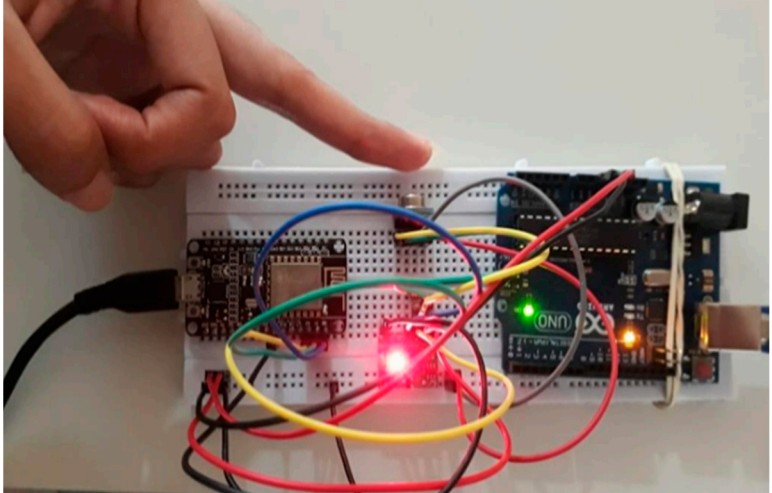

**Figure 4.** Schematization and output of temperature data.

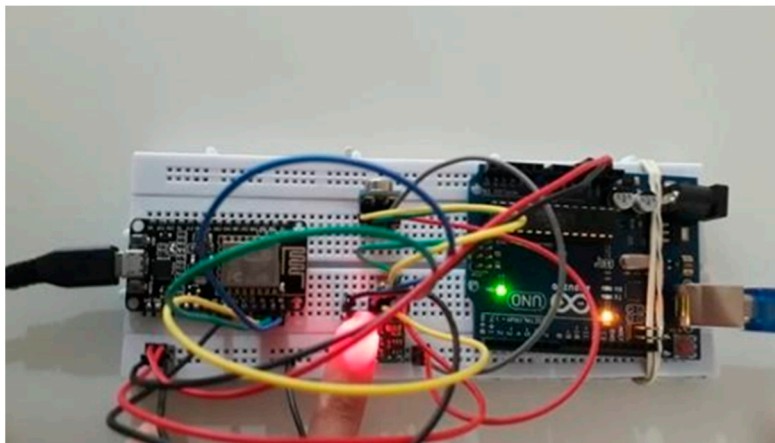

**Figure 5.** Schematization and data output of SpO2 and BPM.

The model used for the implementation of the second sensor is shown in Figure 5, corresponding to the schematic of the Max30100 sensor. The same sensor was in charge of obtaining the values corresponding to oxygenation and heart rate; in addition, the use of this sensor in the first phase was completed, and the second phase was carried out where the previous schematics were joined, giving a final result. The programming and configuration of the prototype were developed and worked on; this information can also be found in the link to the repository hosted on GitHub5. It should be noted that the implemented sensors, especially the temperature sensor, require calibration to eliminate that erroneous information, so it was found that without adequate calibration, the prototype would be very inaccurate. This would be the final result of objective number two, so each sensor works together, visualizing the output data once developed and implemented. To achieve this result, some inconveniences were found. The sensors both work with the same I2C communication line, so when joining them, they did not adapt; the solution for this was to determine the physical addresses of both sensors and thus provide this data to the microcontroller so that the sensors can be differentiated at the time of information request.

*3.5. Use of the Cayanne myDevices Platform*

Based on the comparative analyses of the IoT platforms [15], the Cayenne server was considered the optimal tool to store and monitor the data provided by the sensors from the web; moreover, it is one of the most complete and provides excellent security when transmitting and receiving data using the MQTT protocol. This is the first phase in implementing IoT technologies. Therefore, it must be verified that the hardware has access to the Internet and that the board is recognized by the Cayenne server. To use this server, the CayenneMQTT library is installed, which will allow it to send the physiological parameters. The microcontroller should only incorporate the TCP/IP protocol so that it works with the MQTT protocol; the respective configurations must be made directly from the source code in this case as we use the Esp8266 and therefore, we include the "CayenneMQTTESP8266.h" library. Cayenne Cloud allows you to control any hardware through virtual or analog pins; in this case, we opted for virtual pins, which allow us to display and send the parameters that store the physiological variables that come from the microcontroller and that are received by the Cayenne Cloud server and published in the Cayenne myDevices web dashboard; we define these pins in the source code for sending data. It is worth mentioning that the maximum number of packages that Cayenne can receive is unlimited without the need for a premium account; among some of its limitations that we can find are: that it receives 10 packages per second, 60 packages per minute and 50 connections to the server every 10 min, that is why in the coding a function was assigned so that the data is updated every three seconds.

Finally, the values of temperature, oxygen saturation and heart rate are found; these values are updated and, in turn, stored on the server in real-time; in addition, we determine that both sensors receive the readings corresponding to the electronic prototype and we can verify it by the Serial Monitor. Alert notifications were also added to this research work; to allow this, some configurations were made to the Cayenne server. Cayenne provides the functionality to assign alerts only to the widgets that we want to notify; these alert notifications can be sent via SMS or email; in this case, we configured it for email, and an alert notification was sent to the email address specifying that it has detected a heart rate >100. A comparative analysis of the IoT projects was carried out to measure physiological parameters based on the proposal that was fulfilled, demonstrating their similarities and differences in each aspect defined by the author in this documentation. It was possible to denote that the research works described above do not use many servers in the cloud since most used Bluetooth connectivity.

### 3.6. Blynk App Deployment

After conducting an interview, the medical expert specified that a mobile app would be necessary and useful because users will not always have a laptop at their disposal. An important aspect is that the user can understand the information being provided. For this reason, in this second phase, after comparing IoT applications, the Blynk App was chosen as the optimal tool [16]. This app will be used to visualize and monitor the physiological parameters of the patient; being free, it is easier for the user to acquire. Currently, the Blynk platform has released a new version called Blynk IoT; a new feature of this app is that it can connect to a dashboard on the web, but unfortunately, with it being new, it has some flaws, and when designing the GUI, it is more complicated than expected. That is why the previous version was used; a detailed step by step used guide for how to develop your own GUI and have control of your IoT project in the app was created and made accessible by following a link.

Like the cloud server, it requires a Wi-Fi connection and the use of libraries that authorize it to send physiological parameters, and at the same time, it will provide a token so that the app has a direct connection with the prototype since this is working with the app link (token). It is worth mentioning that during the configuration and design stage of the graphical user interface for the respective monitoring, the pins that are activated and deactivated were selected, for which the virtual pins were used, which are like channels that are in charge of displaying and sending any data from the microcontroller to the app; in addition, it can read devices that use I2C protocols, which is very favorable since the sensors that are being handled use this protocol, as mentioned above.

As mentioned above, the virtual pins were used in conjunction with the physiological variables; this information is sent as strings (a string of text characters) as seen in line 1; the pin is assigned as the first parameter and the data that we want to send to the widget as the second parameter, for this we have concatenated these two variables (oxygen saturation and heart rate) with text strings as labels. Likewise, these data are updated every 3 s. At the end of the design and configuration of the GUI, it will be ready to monitor in real-time the information of physiological parameters through the different widgets that the sensors can connect to.

In Figure 6, the results of the mobile application presenting the data of heart rate, temperature and oxygen saturation are presented. The mobile application works jointly with the previously implemented server, keeping the data updated in real-time; in addition, the two tools will always have a direct connection with the electronic prototype, and thus finally, the patient or user will be able to monitor or have control over their health status from any sector they are in, with this, objective number three is concluded.

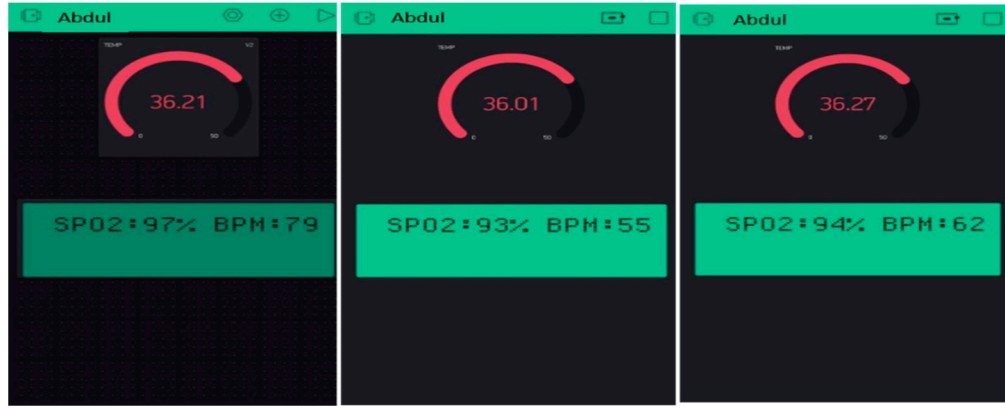

**Figure 6.** Real-time monitoring working in conjunction with the Blynk App.

### 3.7. Electronic Prototype Tests

This stage consists of testing the prototype already implemented and designed to determine the level of acceptance with the readings it provides, comparing it with the commercial devices already available to the doctors using the data collection sheet instrument. In addition, the correct transmission of the data to the Cayenne myDevices platform and the operation of the sensors for real-time monitoring were also verified. Collaboration between teachers was very important for the testing procedure of the electronic prototype. As highlighted, only 10 teachers were chosen for this first phase, aged between 20 and 35 years; in addition, respect and patience were always maintained to complete this process. Help was provided by a nurse who knew how to cooperate during the testing process. Figure 7 corresponds to the temperature results obtained from both the electronic prototype and that of the doctor; in order to verify its precision, the margin of error was handled using the equation of absolute error (2) and relative error (3).

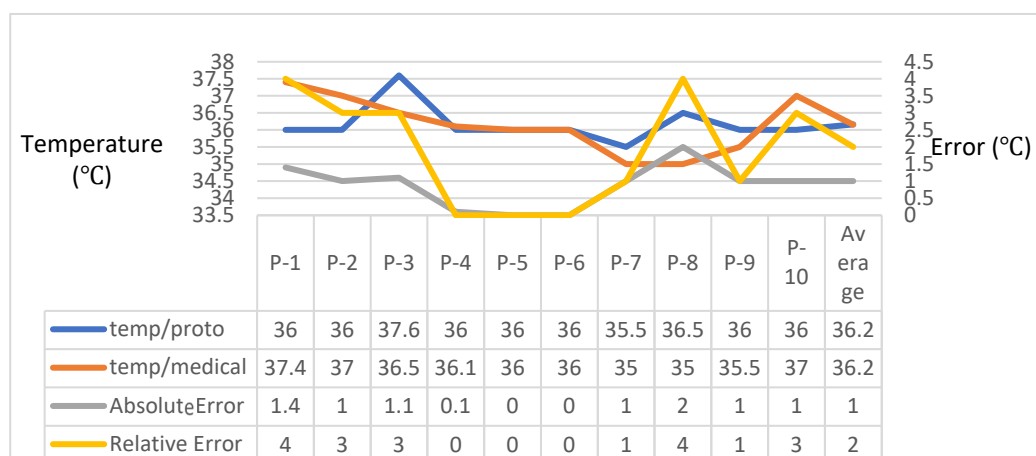

| | P-1 | P-2 | P-3 | P-4 | P-5 | P-6 | P-7 | P-8 | P-9 | P-10 | Average |
|---|---|---|---|---|---|---|---|---|---|---|---|
| temp/proto | 36 | 36 | 37.6 | 36 | 36 | 36 | 35.5 | 36.5 | 36 | 36 | 36.2 |
| temp/medical | 37.4 | 37 | 36.5 | 36.1 | 36 | 36 | 35 | 35 | 35.5 | 37 | 36.2 |
| Absolute Error | 1.4 | 1 | 1.1 | 0.1 | 0 | 0 | 1 | 2 | 1 | 1 | 1 |
| Relative Error | 4 | 3 | 3 | 0 | 0 | 0 | 1 | 4 | 1 | 3 | 2 |

**Figure 7.** First temperature results.

#### 3.7.1. Initial Tests of the Electronic Prototype

Each of the readings is calculated to achieve a global average of the prototype in measuring the temperature parameter, having a degree of inaccuracy of 1% and measurement quality of 2%, which presents a minimum degree of accuracy based on this first factor.

Similarly, Figure 8 shows the heart rate results that, as in the previous case, the margin of error was used through the absolute error, relative error and the equations; in this way, each of the readings is calculated. In order to obtain a global average, this parameter obtained a degree of inaccuracy of 1.2% and measurement quality of 1%, allowing a minimum accuracy to be established regarding this second parameter.

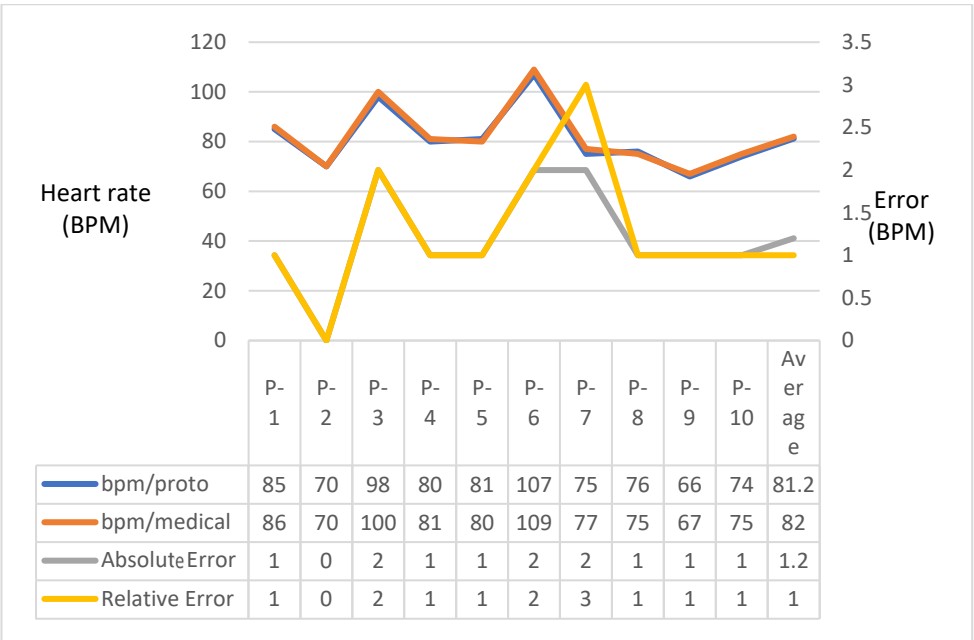

**Figure 8.** First heart rate results.

The results observed in Figure 9 are oxygen saturation, compared with the two factors mentioned above; the imprecision error is higher, giving 2.2% and a measurement quality of 2% that continues to maintain a minimum level in terms of accuracy.

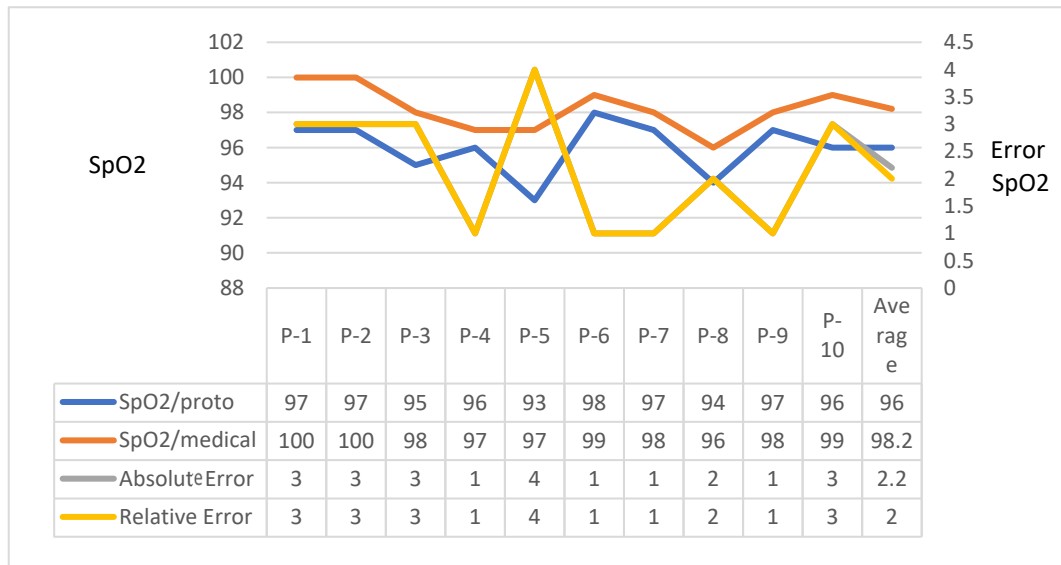

**Figure 9.** Oxygen saturation results.

### 3.7.2. Final Tests of the Electronic Prototype

Final tests of the prototype, according to the sample mentioned above, were carried out with the remaining 17 teachers with an age range between 29 to 69 years. The degree of reliability and accuracy of the data was verified by the electronic prototype, as well as the previous tests; the correct transmission of the data by the server, the application and its OLED screen was taken into account. To certify that the data was not falsified, the final tests of the electronic prototype are highlighted as shown in Figure 10.

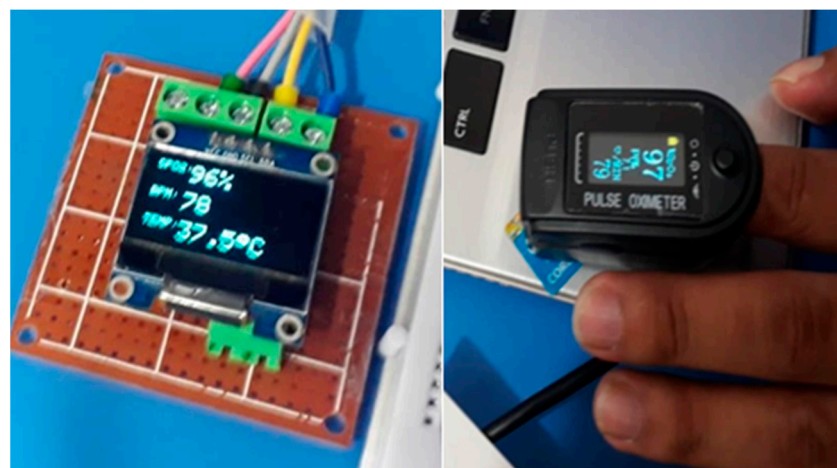

**Figure 10.** Final tests with users.

The results shown in Figure 11 are the temperature values compared between a medical device and the electronic prototype; it is globally determined that the prototype measured with body temperature has an impressive degree of 0.3 °C and measurement quality of 0.8%, which concludes that a minimum error was obtained. It is worth mentioning that the precision is more exact than the first temperature tests, showing that calibrating the sensor reduces its margin of error.

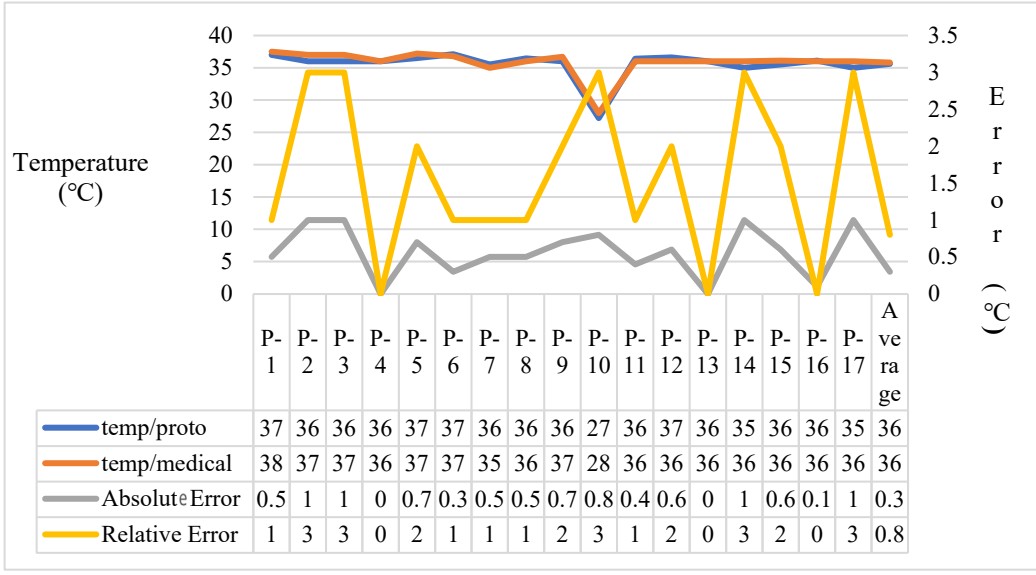

| | P-1 | P-2 | P-3 | P-4 | P-5 | P-6 | P-7 | P-8 | P-9 | P-10 | P-11 | P-12 | P-13 | P-14 | P-15 | P-16 | P-17 | Average |
|---|---|---|---|---|---|---|---|---|---|---|---|---|---|---|---|---|---|---|
| temp/proto | 37 | 36 | 36 | 36 | 37 | 37 | 36 | 36 | 36 | 27 | 36 | 37 | 36 | 35 | 36 | 36 | 35 | 36 |
| temp/medical | 38 | 37 | 37 | 36 | 37 | 37 | 35 | 36 | 37 | 28 | 36 | 36 | 36 | 36 | 36 | 36 | 36 | 36 |
| Absolute Error | 0.5 | 1 | 1 | 0 | 0.7 | 0.3 | 0.5 | 0.5 | 0.7 | 0.8 | 0.4 | 0.6 | 0 | 1 | 0.6 | 0.1 | 1 | 0.3 |
| Relative Error | 1 | 3 | 3 | 0 | 2 | 1 | 1 | 1 | 2 | 3 | 1 | 2 | 0 | 3 | 2 | 0 | 3 | 0.8 |

**Figure 11.** Final body temperature data.

Figure 12 corresponds to the heart rate values (BPM); like in the previous measurement, the margin of error determined by the absolute and relative error is taken into account for this second parameter. Heart rate values reached a lower global error, the degree printing was 1.1%, and measurement quality was 1.5%; this last figure is due to the fact that three samples obtained slightly high records due to various factors, including remaining still or the position of the fingers when taking the measurements. Measurements, being thus the data measured in BPM, are favorable.

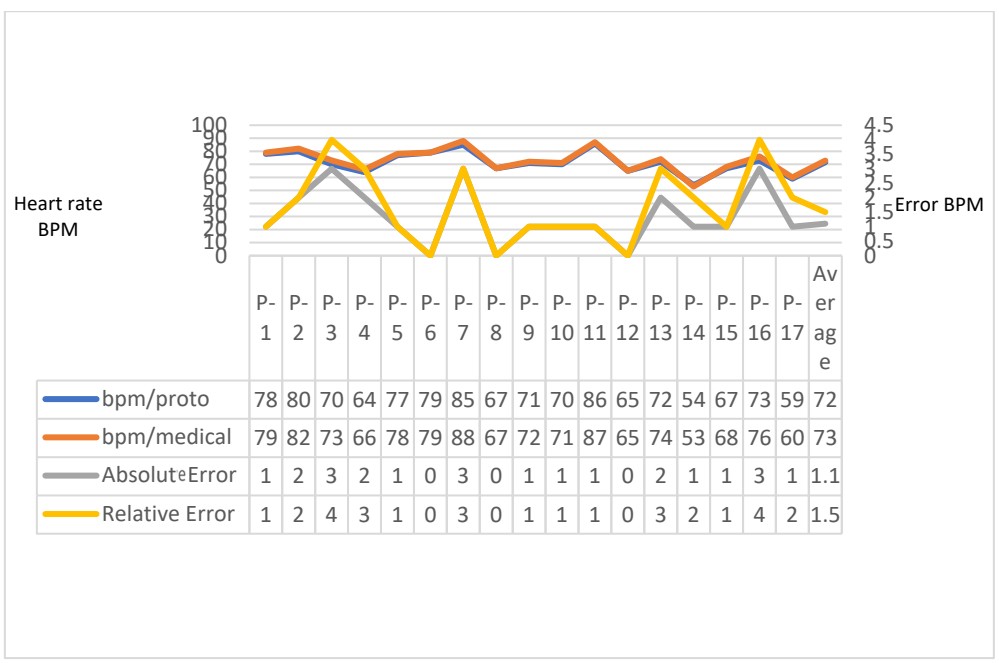

**Figure 12.** Final heart rate data.

Figure 13 shows the records of oxygen saturation (SpO2). The global margin of error for this third minimum parameter was compared to the first tests presented previously; its global impression achieved 0.8% and, as a measure, 0.8%.

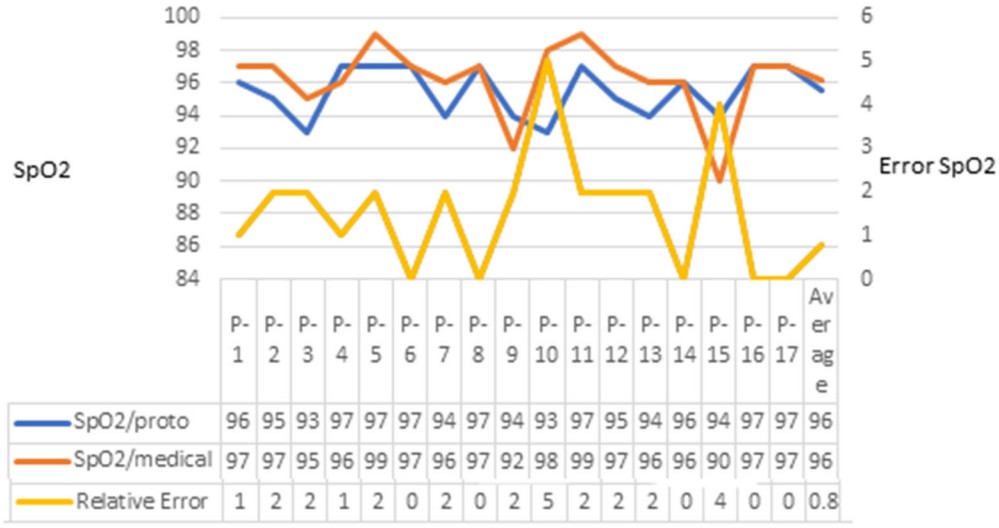

**Figure 13.** Final oxygen saturation data.

*3.8. Electronic Prototype Evaluation Framework*

The evaluation method used in this research was based on certain parameters, and the following characteristics were selected. Next, the parameters to be evaluated were established in a summarized way where the evaluation framework will be located, thus obtaining the quality of this developed prototype according to the criteria for evaluating the electronic prototype as the following:

- Error range
- Response time
- Functionality
- Flexibility
- Resource consumption

Table 3 shows the results concerning the evaluation of the prototype developed and implemented by its author in order to know the quality and satisfaction that this device acquired.

**Table 3.** Results of the evaluation of the electronic prototype (Quality).

| Aspects to Evaluate | Assessment | | | | |
|---|---|---|---|---|---|
| | 1 | 2 | 3 | 4 | 5 |
| Error range | | | | | √ |
| Response time | | | | | √ |
| functionality | | | | | √ |
| Flexibility | | | | | √ |
| Resource consumption | | | | | √ |

Once the results of the evaluation and implementation of the prototype have been presented, the explanation of the results can be decreed. The first parameter to be evaluated was the margin of error, which established a score of less than 5%. The prototype, when subjected to different tests, as previously shown, reached a minimum margin of error for each measured parameter. This margin is equivalent to the temperature parameter, 0.3 °C, heart rate parameter, 1.1%, and oxygen saturation parameter, 0.8%, as shown in Figures 11–13. In the second parameter, the prototype response time from the moment in which the patient makes direct contact with the sensors was evaluated, obtaining a score of five points. The microcontroller takes 15 s to initialize the sensors and the OLED screen; when establishing the I2C protocol, the microcontroller is the one that coordinates the communication with sensors since it is responsible for generating and exchanging data when the patient makes direct non-invasive contact with the sensors. The microcontroller transfers this information to the server and the app, in which, when displayed on the serial monitor, it takes approximately between 2.85 to 3 s to successively read the data until reaching a real value. In addition, let us remember that these data vary and that the information that is sent to the server and the app takes exactly the same seconds as mentioned above. Moreover, this can vary as long as the Wi-Fi network connection does not oscillate in a high status.

The third parameter, functionality, was evaluated and obtained a score of five. The prototype meets all the requirements that correspond to this research and works optimally and beneficially, which was validated by a medical professional who determined that it is very close to the optimal performance of equipment designed to obtain vital signs. The fourth parameter, flexibility, stands out, as it obtained a score of five. These results were obtained after analyzing and determining that the prototype exchanges direct information with the user by monitoring their physiological parameters, assuming that their processes are in real-time and update every 3 s. Finally, the last parameter to be evaluated was the consumption of resources, which obtained a score of five because the more resources the computer has when it is executed, the more the prototype tends to take less time, and communication is much faster since this device tends to have 125 times more memory than the ATmega328.

### 3.9. Assessment Framework according to ISO/IEC 25012

This evaluation framework used in this research is based on the parameters of ISO/IEC 25012. It is a summarized form of the parameters to be evaluated based on the standard ISO/IEC 25012 in order to determine the quality of the data provided by the electronic prototype. To better understand the results, ranges of values classified as Unacceptable, Minimally Acceptable, and Target Range were implemented for each parameter, as shown in Table 4.

**Table 4.** Decision criteria for the body temperature, heart rate, and oxygen saturation parameters.

| | | |
|---|---|---|
| Body temperature | Unacceptable | If Value > 3 |
| | minimally acceptable | If Value $\geq$ 1 and Value $\leq$ 3 |
| | target range | If Value $\geq$ 0, 1 and Value < 1 |
| heart rate | Unacceptable | If Value > 5 |
| | minimally acceptable | If Value > 1 and Value $\leq$ 5 |
| | target range | If Value > 0.1 and Value $\leq$ 3 |
| Oxygen saturation | Unacceptable | If Value > 5 |
| | minimally acceptable | If Value > 1 and Value $\leq$ 5 |
| | target range | If Value > 0.1 and Value $\leq$ 3 |

Table 5 shows the results of the data evaluation according to the ISO/IEC 25012 standard to specify the integrity of the data of the electronic prototype.

**Table 5.** Evaluation results for the temperature parameter, heart rate and oxygen saturation tests (Assessment according to ISO/IEC 25012).

| Characteristic | Aspects to Evaluate | Decision Criteria | | |
|---|---|---|---|---|
| | | Unacceptable | Minimally Acceptable | Target Range |
| Inherent data quality | Accuracy | | | √ |
| | Consistency | | | √ |
| | Credibility | | √ | |
| System-dependent data quality | Accessibility | | | √ |
| | Efficiency | | | √ |
| | Precision | | | √ |
| | understandability | | | √ |
| | Availability | | | √ |

After presenting the evaluation results for the temperature parameter from Table 5 using scientific observation, the foundation for the results displayed can be established. Accuracy, consistency, and credibility were evaluated in the first section, which dealt with the inherent data quality. When tabulating the final tests, the first parameter, accuracy, produced an error of less than 1 °C, demonstrating that the data becomes nearly exact when compared to a medical one. As a result, it reached the objective range. The second parameter, consistency, was within the target range because the patient could reasonably visualize their data and determine whether their health was good or if there was an anomaly in his body. The prototype became at least minimally acceptable as a third parameter for credibility because of the prototype's small margin of error and the accuracy of the data.

According to the system-dependent data quality, five factors were evaluated in the second section: accessibility, Efficiency, accuracy, comprehension, and availability. The first accessibility parameter was met because the user can always access their physiological parameter data whether or not they have Internet access. Because it was efficiently maximized and only took a few seconds compared to the time needed to display the data, the second parameter, Efficiency, also achieved its goals. The Precision, Comprehension, and Availability parameters, whose information is accurate in some circumstances and close to the real value in others, finally achieved the objective range. Additionally, by having the user use a mobile application, this research work enables you to always have access to your information. After presenting the evaluation results for the heart rate parameter from Table 5 using scientific observation, the foundation for the results displayed can be established. Accuracy, consistency, and credibility were evaluated in the first section, which

dealt with the inherent data quality. When the final tests were tabulated, the heart rate was off by 1%, and the oxygen saturation was off by 0.8%, demonstrating that the data was nearly accurate when compared to a doctor, so the first parameter, accuracy, was within the desired range. The second parameter, consistency, was within the target range because the patient could reasonably visualize their data and determine whether his health was good or if there was an anomaly in his body. The prototype became at least minimally acceptable as a third parameter for credibility because of the prototype's small margin of error and the accuracy of the data. According to the system-dependent data quality, five factors were evaluated in the second section: accessibility, Efficiency, accuracy, comprehension, and availability. The first accessibility parameter was met because users can always access their physiological parameter data whether or not they have Internet access. Efficiency, the second parameter, is effectively maximized because it falls within the target range, given that it only takes a few seconds to display the data.

The objective range was finally attained with the Precision, Comprehension, and Availability parameter because the data it provides is sometimes accurate and other times close to the true value. Additionally, this research employs a mobile application that the user can use to dispose of their information and permit it to be tracked for however long they require.

## 4. Discussion

In this study, we discovered that Trivedi and Cheeran, 2018 [17], created a device to monitor physiological parameters. However, the methods used were different, both in the use of sensors and microprocessors and even in the way data was shared with the prototype since they used Wi-Fi connectivity because the microcontroller they incorporated had a Wi-Fi module integrated. Compared to the study by Castellanos, Guadalupe, Reyes, and Sanchez [18], which was not exact because of their patients' mobility, the research results were good because they met the desired objective and had a small margin of data error.

According to the findings above, regarding the calculated errors, the first parameter evaluated was the margin of error, which achieved a target score of less than 5% due to the prototype's performance during various tests. As previously demonstrated, the prototype reached a minimum margin of error for each measured parameter. These parameters were temperature (with an error of 0.3 °C), heart rate (with an error of 1.1%), and oxygen saturation (with an error of 0.8%), as depicted in Figures 11–13.

Another study that used a similar technique, Rodrguez, Postolache, and Cercasc [7], measured physiological parameters like BPM and photoplethysmography that serve to measure changes in the user's environment, implementing sensors inside office chairs, wheels, and beds, which does not make it useful in a portable way because it must always remain in a fixed place, for this reason, it was decided to develop a prototype.

Another disadvantage of the study is the use of Bluetooth for data connectivity due to the limitation of distance and the possibility of measurement errors even at the ideal distance. This is in contrast to Wi-Fi connectivity, where distance is not a disadvantage, and you can take physiological parameters using an IoT application connected to it from anywhere. Unlike the author, who used a light-dependent sensor with various implementations, this research, as previously mentioned, used the Max30100 sensor to measure BPM and SpO2 in such a way that it does not require specific lighting conditions to take the corresponding readings.

A high level of positivism was achieved when the research was analyzed in the prototype's usability section, as opposed to Murali (2018) [9], who did not succeed because they created a wireless belt that the patient must constantly put on and that causes discomfort.

It should be noted that there were several failures in the results obtained in this study, with a margin of error of 25%, which were not valid results and thus not accepted in the medical field; in addition, one of the mistakes they made was that they were not based on using IoT platforms generating more effect. The study proposed by Revar [19] aims to create a pulse oximeter that measures heart rate and oxygen saturation, preventing respiratory or

cardiac diseases. Another thing to keep in mind is that most of the presented works use Bluetooth as a data transmission route and that very few studies use IoT platforms.

## 5. Conclusions and Recommendations

### 5.1. Conclusions

There are an infinite number of microcontrollers and sensors to create electronic proto-types, particularly for the medical industry that measures physiological parameters. To avoid giving the patient inaccurate results, we must carefully select the instruments, so they were assessed to ensure a 95% effectiveness level. All of the free software tools used in this research, such as Cayenne myDevices (cloud storage), Blynk App (graphical user interface and real-time visualization), and Arduino (coding), allow for greater integration of changes and processes and reduce the need for expensive applications and IoT platforms that perform the same function. Patients with COVID-19, cardiovascular patients, diabetics, and others can use an electronic prototype for monitoring physiological parameters such as temperature, heart rate, and oxygen saturation while exercising. A successful prototype was produced using the Max30100, Mlx90614 sensors, and Esp8266 microcontroller to enable data transmission and visualization. Additionally, the temperature sensor's technical capabilities were insufficient to accurately measure a human's body temperature, necessitating a calibration algorithm. The Cayenne myDevices platform offers a web dashboard that continuously tracks physiological parameters and stores this data, making it simple for the patient to access and browse a daily log. Data from the server and the electronic prototype are visualized and tracked in real-time by Blynk, an IoT-based application with a graphical user interface. Designing widgets and configurations was simple with Blynk's interface. Surveys revealed that the usability and data quality of the electronic prototype was very high. Response times were quick, but during the final testing, problems with the internet connection prevented the platform and app from communicating with the electronic prototype. Due to excessive finger movement and posture, some measurements were taken at the upper limit of the acceptable range.

### 5.2. Recommendations

This research will lead to advances similar to or even better than the one that was presented, especially by creating new development opportunities in Esmeraldas and showing them how to advance the technologies in their environment. It would help to have a clear idea of what you want to accomplish when creating an IoT-based research project and be well-versed in the operation of each sensor through data shed readings, which will reveal its voltage, technical support, input type, and microcontroller compatibility. If not, problems could develop later on, and the desired outcome might not be realized. Antistatic boxes can shield your device from static electricity when working with electrical devices. Be careful not to burn sensitive sensors. To obtain an accurate reading of a physiological parameter, the prototype must be in a fixed location, and the patient must be at ease. Error is increased by movement and anxiety. When used in the real world, this device completely satisfies the sample, so research needs to be improved. Sensors could not be added to increase accuracy because they are difficult to come by in Libya and are only available to medical teams.

**Author Contributions:** Conceptualization, S.J. and A.A.; methodology, A.A.; software, S.J.; validation, A.A.; formal analysis, S.J.; investigation, A.A. and S.J.; resources, A.A.; data curation, S.J. and A.A.; writing—original draft preparation, A.A.; writing—review and editing, S.J.; visualization, S.J.; supervision, S.J. and A.A.; project administration, A.A. and S.J.; All authors have read and agreed to the published version of the manuscript.

**Funding:** This research received no external funding.

**Institutional Review Board Statement:** The study was approved by Karpas Mediterranean University of Medicine Ethics Committee. Informed patient consent was not required by the ethics

committee in view of the retrospective nature of the research and the anonymity of the study data. The guidelines outlined in the Declaration of Helsinki were followed.

**Informed Consent Statement:** Informed consent was obtained from all subjects involved in the study.

**Data Availability Statement:** The data used to support the findings of this study are included in the article.

**Conflicts of Interest:** The authors declare no conflict of interest.

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
