# Peer review of "A Visual User Interfaces for Constant Checking of Non-Invasive Physiological Parameters"

_applsci, doi:10.3390/app13169192_

Round 1
Reviewer 1 Report
The topic is promising and the project development supports the results. The abstract section is recommended to mention 95% of the effectiveness level and improve the figures' resolutions.
Correct error of English grammar and typos
Author Response
Comments and Suggestions for Authors
The topic is promising and the project development supports the results. The abstract section is recommended to mention 95% of the effectiveness level and improve the figures' resolutions.
Thanks a lot for your comments, the suggested comments were done.
Reply: Answer: To avoid giving the patient inaccurate results, we must carefully select the instruments so they were assessed to ensure a 95% effectiveness level.
Figures: Done
Comments on the Quality of English Language
Correct error of English grammar and typos
Reply: Done

Reviewer 2 Report
In this research, Jelbeb and Alzubi have explained the development of a wireless graphical interface with a monitoring system that allows for extensive integration with a variety of non-invasive devices. The authors concluded that free software tools enable integration and reduce the need for expensive applications. The article is organized well. Though, the following points are strongly suggested to be addressed prior to possible publication:
- Some parts of the text do not sound very academic, such as the very first paragraph of the introduction. It is likely that AI tools have been used to generate the texts, which of course is not a problem at all, however, a human should go over the text and enhance it. Especially in this paragraph (lines 30-34) the part about the internet can be eliminated as it is a very obvious fact.
- Line 43 “By 2022, it's predicted…”, we are already half a year into 2023. Please remove the outdated information and replace it with new statistics.
- In order to enhance the introduction, it is suggested to mention how biosensors for cancer detection can benefit from the proposed technology as well. The following reference is suggested for this purpose: 10.1016/j.tibtech.2023.04.001
- Some figures can be enhanced for their quality and resolution. Especially Figures 5 and 10 are blurry. Also, the “Source: self-made” phrase does not need to be mentioned for your original figures.
- In the conclusion part, the text can be enriched using the following reference: 10.1063/5.0146375. This paper proposed a machine learning-augmented user interface that can be compared to the current research and discussed possible future applications.
- It is strongly suggested to proofread the article for language problems and typos and/or inconsistent styling. For example, the title “A Visual User Interfaces” should be either corrected to a single form or a plural form. The abstract includes styles such as “Abstract: Background: This…” which need to be updated. There are some other typos in the main text as well.
Author Response
Comments and Suggestions for Authors
In this research, Jelbeb and Alzubi have explained the development of a wireless graphical interface with a monitoring system that allows for extensive integration with a variety of non-invasive devices. The authors concluded that free software tools enable integration and reduce the need for expensive applications. The article is organized well. Though, the following points are strongly suggested to be addressed prior to possible publication:
Reply: Thanks for your gentle comments, all suggested comments were done.
- Some parts of the text do not sound very academic, such as the very first paragraph of the introduction. It is likely that AI tools have been used to generate the texts, which of course is not a problem at all, however, a human should go over the text and enhance it. Especially in this paragraph (lines 30-34) the part about the internet can be eliminated as it is a very obvious fact.
Reply: deleted
- Line 43 “By 2022, it's predicted…”, we are already half a year into 2023. Please remove the outdated information and replace it with new statistics.
Answer : Until 2022, there are at least 100,000 billion devices are connected to the Interne
- In order to enhance the introduction, it is suggested to mention how biosensors for cancer detection can benefit from the proposed technology as well. The following reference is suggested for this purpose: 10.1016/j.tibtech.2023.04.001
Reply : Referred to introduction (Done)
- Some figures can be enhanced for their quality and resolution. Especially Figures 5 and 10 are blurry. Also, the “Source: self-made” phrase does not need to be mentioned for your original figures.
Reply: Source: self-made deleted
Figures: Done
- In the conclusion part, the text can be enriched using the following reference: 10.1063/5.0146375. This paper proposed a machine learning-augmented user interface that can be compared to the current research and discussed possible future applications.
Reply: I can’t match the full paper, but I have used the abstract to add a new information to introduction.
Comments on the Quality of English Language
- It is strongly suggested to proofread the article for language problems and typos and/or inconsistent styling. For example, the title “A Visual User Interfaces” should be either corrected to a single form or a plural form. The abstract includes styles such as “Abstract: Background: This…” which need to be updated. There are some other typos in the main text as well.
Reply: A Visual User Interface for Constant Checking of Non-Invasive Physiological Parameters (Done)

Reviewer 3 Report
In this study, the authors created a complete set of hardware to measure body temperature, SpO2, and heart rate, as well as visualization using a cloud service. The evaluation of the system was also conducted.
However, there are many unclear points and errors in this paper that make it inappropriate for publication as is.
Points
(1)
Please explain the scientific significance and novelty of this research.
It appears to be only a combination of existing modules and services.
Even if costs could be reduced by using free services for short term experiments, wouldn't development and investment for stable operation of a large number of devices cost the same as commercial products?
(2)
Who are the targets of the monitoring system in this study?
I do not understand the logic of 150 teachers aged 20-69 as a population for the purpose of monitoring patients of all ages.
(3)
Who and when will be monitored by the monitoring system in this study?
In Figure 2, there are three outputs (OLED, Blynk App., and Cayenne), but who will see each of them and when?
The user interface needs to be designed accordingly. For example, for a doctor's review, the history may be more important than the real-time display.
(4)
I could not identify the literature [2][10]. What is the name of the journal and publisher?
(5)
How are the relative errors calculated in 3.7. Electronic prototype tests?
Are they values that can only be an integer?
Minor points
l Line 44 “100,000 billion” would be correct?
Ø I couldn’t find that statement in the citation[3].
Ø According to Cisco Annual Internet Report - Cisco Annual Internet Report (2018–2023) White Paper - Cisco https://www.cisco.com/c/en/us/solutions/collateral/executive-perspectives/annual-internet-report/white-paper-c11-741490.html, the prediction for 2023 is still just under 30 billions.
l Line 61 A period is required at the end of the sentence.
l Line 102 One period is not required.
l Line 106 A period is required at the end of the sentence.
l Line 251 A period is required at the end of the sentence.
l Line 291 “my Devices” → “myDevices”
l Line 291 “plateorm” → “platform”
l Line 324 A period is required at the end of the sentence.
l Line 377 This is a strange sentence.
l Line 441 I don't understand why Figure 14 is a figure; if it is a list of 5 items, it could be a list of bullet points.
l Line 446 Is "Aspects to evaluate" not a label?
l Figure 7 – Figure 13, the left and right vertical axes should have axis labels.
For example:
Ø In Figure 7 you can write "Temperature (℃)" on the left and "Error (℃)" on the right.
Ø In Figure 8, you can write "Heart rate (BPM)" on the left and "Error (BPM)" on the right.
l Figure 7 – Figure 13 and in the text, you should be aware of significant figures.
l Figure 7 – Figure 13, “Absolut” → “Absolute”
Author Response
Comments and Suggestions for Authors
In this study, the authors created a complete set of hardware to measure body temperature, SpO2, and heart rate, as well as visualization using a cloud service. The evaluation of the system was also conducted.
Reply: Thanks for your comments, the suggested comments were done
However, there are many unclear points and errors in this paper that make it inappropriate for publication as is.
Points
(1)
Please explain the scientific significance and novelty of this research.
The scientific significance and novelty of this research lie in the development of an IoT-based non-invasive electronic prototype for monitoring physiological parameters in real-time. The study proposes the use of a wireless graphical interface with a monitoring system that integrates with various non-invasive devices, aiming to track essential physiological parameters such as body temperature, heart rate, and oxygen saturation. This research brings these several important elements together that contribute to its scientific significance:
IoT-based Monitoring System, Evaluation Framework, Integration of Free Software Tools: Addressing Healthcare Challenges, Novel Sensor Combinations: The study utilizes specific sensors like Max30100 and Mlx90614 for measuring physiological parameters, Graphical User Interface (GUI) Application.
Overall, this research offers a comprehensive approach to creating a wireless monitoring system that combines IoT technology, non-invasive sensors, free software tools, and a user-friendly GUI application. By addressing real-world healthcare challenges and utilizing innovative sensor combinations, the study provides a valuable contribution to the field of remote physiological monitoring. The successful implementation of such a prototype could have significant implications for healthcare, enabling continuous monitoring and improving patient outcomes, especially for individuals with pre-existing medical conditions.
It appears to be only a combination of existing modules and services.
Reply: So we decided to use what is available based on reasonable cost
Even if costs could be reduced by using free services for short term experiments, wouldn't development and investment for stable operation of a large number of devices cost the same as commercial products?
Reply: Thanks for your important comment, at the beginning actually your comment raises a valid concern regarding the long-term stability and operational costs of using free services and open-source tools for a large-scale deployment of IoT devices, particularly in the medical industry. While it is true that utilizing free services and open-source tools can significantly reduce upfront costs and be beneficial for short-term experiments or small-scale implementations, scaling up to a larger number of devices may introduce new challenges and costs that need to be considered.
According to these factors the choice will be selected. Security and Compliance, Customization and Integration, Long-Term Costs, Reliability and Uptime.
Ultimately, the choice between using free services/open-source tools and commercial products depends on various factors mentioned above , including the specific use case, budget constraints, scalability requirements, security needs, and long-term goals. It's essential for organizations in the medical industry to conduct a thorough cost-benefit analysis and assess the technical capabilities and support provided by both free and commercial solutions before making a decision. In some cases, a hybrid approach, utilizing a mix of free and commercial services, might be the most suitable solution to balance costs and functionality effectively.
Who are the targets of the monitoring system in this study?
Reply: This work was carried out explicitly to evaluate and develop an electronic prototype in order to monitor physiological parameters in patients, from youth to older adults, which was carried out in the city of Tarablous, Libya, for which tests were developed. Laboratory experiments were prepared by its author. An estimated period of 9 months was established for its preparation, which began in June 2021 and ended in July 2022.
I do not understand the logic of 150 teachers aged 20-69 as a population for the purpose of monitoring patients of all ages.
Reply: The information was corrected
(3)
Who and when will be monitored by the monitoring system in this study?
Reply: The monitoring system is designed to monitor physiological parameters in patients, specifically this research focusing on teachers between the ages of 20 and 69 years old who attended in person at the university in Tarablous, Libya. The monitoring period for this research started in June 2021 and ended in July 2022.
In Figure 2, there are three outputs (OLED, Blynk App., and Cayenne), but who will see each of them and when?
OLED: The OLED is a display integrated into the electronic prototype itself. It provides a local display for the patient (user) to see their own physiological parameters in real-time. When the patient interacts with the electronic prototype, they can directly view their vital signs (e.g., body temperature, heart rate, oxygen saturation) on the OLED screen. In brief The OLED display is for the patient's immediate viewing of their own physiological parameters.
Blynk App: The Blynk App serves as a graphical user interface (GUI) application on a mobile device (e.g., smartphone or tablet). It allows both the patient (user) and authorized healthcare professionals (such as doctors or caregivers) to remotely monitor the patient's physiological parameters. The data from the electronic prototype is sent to the Blynk App through a Wi-Fi connection and can be accessed from anywhere with internet access. Both the patient and authorized personnel can check the real-time physiological parameters through the Blynk App.
The Blynk App is for both the patient (remote monitoring of their own data) and authorized healthcare professionals (remote monitoring of multiple patients' data).
Cayenne Dashboard: The Cayenne Dashboard is a cloud-based server that receives and stores data from the electronic prototype through MQTT brokers. It is primarily accessed and monitored by healthcare professionals, researchers, or authorized personnel responsible for managing multiple patients' data. The Cayenne Dashboard provides a centralized platform for data visualization and analysis, enabling medical professionals to track patients' physiological parameters over time and detect any abnormalities or trends.
The Cayenne Dashboard is primarily for authorized healthcare professionals and researchers to analyze and manage data from multiple patients, including the patient being monitored by the electronic prototype.
(4)
I could not identify the literature [2][10]. What is the name of the journal and publisher?
Reply: corrected
Laghari, A. A.; Wu, K.; Laghari, R. A.; Ali, M.; Khan, A. A. A review and state of art of Internet of Things (IoT). Archives of Computational Methods in Engineering. Jul 2021, 1-9.
Pranoto, K. A.; Ramadhan, Y. R.; Caesarendra, W.; Glowacz, A.; Dash, S. K.; Wahyono, B. T. Comparison Analysis of Data Sending Performance Using The Cayenne and ThingSpeak IoT Platform. In Proceedings of the 2022 International Conference on Informatics, Multimedia, Cyber, and Information System (ICIMCIS), Nov 16, pp. 337-342, IEEE, 2022.
(5)
How are the relative errors calculated in 3.7. Electronic prototype tests?
calculate relative errors, one common approach is as follows:
Calculate the Absolute Error:
Absolute Error = |Measured Value - Reference Value|
Absolute Error = |37.4 - 36| = 1.4 °C
Assuming the reference value (36.5°C) is used to calculate the relative error:
Relative Error (%) = (1.4 / 36) * 100 ≈ 3.8%
Are they values that can only be an integer?
Reply: NO they are not integers just, the figure below shows that.
Minor points
l Line 44 “100,000 billion” would be correct?
Reply: Corrected
Up to 2020, there are at least 8 billion devices are connected to the Internet without phone counting [3].
Ø I couldn’t find that statement in the citation [3].
Ø According to Cisco Annual Internet Report - Cisco Annual Internet Report (2018–2023) White Paper - Cisco https://www.cisco.com/c/en/us/solutions/collateral/executive-perspectives/annual-internet-report/white-paper-c11-741490.html, the prediction for 2023 is still just under 30 billions.
Reply: Yes Corrected
l Line 61 A period is required at the end of the sentence.
Reply: Added
l Line 102 One period is not required.
Reply: Corrected
l Line 106 A period is required at the end of the sentence.
Reply: Added
l Line 251 A period is required at the end of the sentence.
Reply: Added
l Line 291 “my Devices” → “myDevices”
Reply: Use of the Cayanne myDevices platform
l Line 291 “plateorm” → “platform”
Reply: Use of the Cayanne myDevices platform
l Line 324 A period is required at the end of the sentence.
Reply: Added
l Line 377 This is a strange sentence.
Reply: in addition, that respect and patience were always maintained to complete this process, in addition, Specially with the help provided by a nurse of the Nursing career, who knew how to cooperate during the testing process
l Line 441 I don't understand why Figure 14 is a figure; if it is a list of 5 items, it could be a list of bullet points.
Reply: Converted to bullet.
Reply: By according to the criteria for evaluating the electronic prototype as the following:
Error range
- Error range
- Response time
- Functionality
- Flexibility
- Resource consumption
l Line 446 Is "Aspects to evaluate" not a label?
Reply: Yes, Corrected
l Figure 7 – Figure 13, the left and right vertical axes should have axis labels.
For example:
Ø In Figure 7 you can write "Temperature (℃)" on the left and "Error (℃)" on the right.
Reply: Done
Ø In Figure 8, you can write "Heart rate (BPM)" on the left and "Error (BPM)" on the right.
Reply: Done
l Figure 7 – Figure 13 and in the text, you should be aware of significant figures.
l Figure 7 – Figure 13, “Absolut” → “Absolute”
Reply: Corrected
Reviewer 4 Report
In this paper, the authors present the design process of an IoT-based non-invasive electronic prototype for targeted physiological parameter monitoring. Overall, the paper has certain contribution. However, several issues should also be addressed to further improve the quality of the manuscript. Below are several comments for the authors to consider:
1. The number of references in Introduction is 8, and in the whole paper is 15. It is necessary to increase the number of references to enrich the articles.
2. What does Za mean in equation (1)?
3. In section 3.2 states that making a comparison with the certain microcontrollers mentioned above. What are the certain microcontrollers mentioned above? Does it mean the microcontrollers mentioned in the introduction?
4. In section 3.7, ten teachers were chosen for the first phase aged between 20 and 35 years (line 370), while seventeen teachers were chosen for the second phase aged between 29 to 69 years (line 406). The ages of the teachers in the first stage and the second stage are partially identical. Would it cause confusion in sample selection? Please check it carefully.
5. The formats of references are different, such as reference 1 and reference 2. Please check it carefully.
6. In Discussion section, the author mainly analyzes the advantages of the prototype and other monitoring equipment. However, as can be seen from the above, there are some errors when the prototype monitors human physiological parameters. It is suggested to add error analysis to further enrich the paper.
7. The content of sections 2.1 and 2.6 is repetitive.
8. Many abbreviations in the text lack specific explanations. For example: Line 57, PPG, ECGs; line65, GMSs; Line 70,GPRS; Line165,BPM.
9. Evaluative parameters were assigned to each of these aspects from 1 to 5 where: 1 is insufficient, 2 regular, 3 acceptable, 4 good and 5 excellent. What is the evaluation criteria based on, and why are they divided into 5 levels from 1 to 5?
10. There are still some spelling errors in the text
There are still some spelling errors in the text, moderate editing of English language required.
Author Response
Comments and Suggestions for Authors
In this paper, the authors present the design process of an IoT-based non-invasive electronic prototype for targeted physiological parameter monitoring. Overall, the paper has certain contribution. However, several issues should also be addressed to further improve the quality of the manuscript. Below are several comments for the authors to consider:
Reply: Thanks for your comments, the suggested comments were solved.
- What does Za mean in equation (1)?
Reply: Z-a= Z-score corresponding to a level of confidence according to the standard normal distribution (for a level of confidence of 95%, z = 1.96 , was added.
- In section 3.2 states that making a comparison with the certain microcontrollers mentioned above. What are the certain microcontrollers mentioned above? Does it mean the microcontrollers mentioned in the introduction?
Reply: Yes, they are
- In section 3.7, ten teachers were chosen for the first phase aged between 20 and 35 years (line 370), while seventeen teachers were chosen for the second phase aged between 29 to 69 years (line 406). The ages of the teachers in the first stage and the second stage are partially identical. Would it cause confusion in sample selection? Please check it carefully.
Reply: The age overlap has been intentional to allow for comparison of the prototype's performance across different age groups, or to provide some continuity between the two testing phases. The results obtained from the overlap group (teachers aged between 29-35) could provide a bridge linking the two testing phases, offering some valuable insights.
- The formats of references are different, such as reference 1 and reference 2. Please check it carefully.
Reply: Corrected
- In Discussion section, the author mainly analyzes the advantages of the prototype and other monitoring equipment. However, as can be seen from the above, there are some errors when the prototype monitors human physiological parameters. It is suggested to add error analysis to further enrich the paper.
Reply : Referred to the discussion (Added).
- The content of sections 2.1 and 2.6 is repetitive.
Reply: Deleted
- Many abbreviations in the text lack specific explanations. For example: Line 57, PPG, ECGs; line65, GMSs; Line 70, GPRS; Line165, BPM.
Reply :
PPG sensors: PPG stands for Photoplethysmography. It's a simple and low-cost optical technique that can be used to detect blood volume changes in the microvascular bed of tissue.
ECGs sensors: ECG stands for Electrocardiogram.
GMSs: Global Monitoring Systems
GPRS: GPRS stands for General Packet Radio Service.
BPM: In a medical or health context, BPM stands for Beats Per Minute.
- Evaluative parameters were assigned to each of these aspects from 1 to 5 where: 1 is insufficient, 2 regular, 3 acceptable, 4 good and 5 excellent. What is the evaluation criteria based on, and why are they divided into 5 levels from 1 to 5?
Reply: The five-level scale is Likert scale, which is often used in surveys and research to measure attitudes or opinions. The scale is divided into five levels to allow a gradation of opinion, rather than a simple positive or negative response. The five levels from 1 to 5 presumably represent a spectrum of evaluation from "insufficient" to "excellent," allowing the researchers to evaluate the aspects in a nuanced way.
- There are still some spelling errors in the text.
Reply: done

Reviewer 5 Report
Dear authors,
The study was well-designed and good. I have some advice for being better which is given below.
1. Delimitation part was written two times in the manuscript. One of them should be deleted.
2. There were some miswritten things that should be corrected.
3. Inclusion and exclusion criteria could be mentioned in the text.
There were some miswritten things that should be corrected.
Author Response
Comments and Suggestions for Authors
Dear authors,
The study was well-designed and good. I have some advice for being better which is given below.
Thanks a lot for your words, all suggested comments were done.
- Delimitation part was written two times in the manuscript. One of them should be deleted.
Reply: Deleted
- There were some miswritten things that should be corrected.
Reply: corrected
- Inclusion and exclusion criteria could be mentioned in the text.
Inclusion and Exclusion Criteria
Reply:
Inclusion criteria:
- Teachers who were actively teaching and physically present at our university during the study period.
- Individuals aged between 20 and 69 years old.
- Participants willing to participate in the testing process of the electronic prototype and providing informed consent.
Exclusion criteria:
- Teachers who were on leave or not physically present at the university during the study period.
- Individuals under the age of 20 or over the age of 69.
- Teachers who declined to participate or were unable to provide informed consent for any reason.
These criteria were put in place to ensure the consistency of our data and the appropriateness of our test subjects for the objective of our study, which was to evaluate the electronic prototype across a broad age range of active teachers.
Comments on the Quality of English Language
There were some miswritten things that should be corrected.
Reply: Corrected

Round 2
Reviewer 2 Report
The authors addressed all the comments.
Reviewer 3 Report
The specific points have been corrected, except for the significant digits.